# Cardiopulmonary Exercise Testing in Childhood in Late Preterms: Comparison to Early Preterms and Term-Born Controls

**DOI:** 10.3390/jpm12101547

**Published:** 2022-09-20

**Authors:** Ori Hochwald, Lea Bentur, Yara Haddad, Moneera Hanna, Merav Zucker-Toledano, Gur Mainzer, Julie Haddad, Michal Gur, Liron Borenstein-Levin, Amir Kugelman, Ronen Bar-Yoseph

**Affiliations:** 1Neonatal Intensive Care Unit, Ruth Children’s Hospital, Rambam Health Care Center, Technion Faculty of Medicine, Haifa 3200003, Israel; 2Pediatric Pulmonary Institute, Ruth Children’s Hospital, Rambam Health Care Center, Technion Faculty of Medicine, Haifa 3200003, Israel; 3Technion Faculty of Medicine, Haifa 3200003, Israel; 4Pediatric Cardiology Institute, Ruth Children’s Hospital, Rambam Health Care Campus, Haifa 3109601, Israel

**Keywords:** late preterms, cardiopulmonary exercise testing, exercise capacity, oxygen uptake

## Abstract

Background: Late preterm (34^0^–36^6^ weeks gestational age [GA]) infants may have abnormal pulmonary development and possible exercise physiology parameters. We aim to assess the effect of late prematurity on exercise capacity in childhood and to compare it to early preterm (EP) (born < 30^0^ GA), and to term healthy control (TC) (>37^0^ week GA). Methods: Late preterm and early preterm (7–10 years) completed a cardiopulmonary exercise test (CPET) and spirometry and were compared to EP and to TC. Results: Eighty-four children (age 9.6 ± 1.0 years, 48% girls) participated. Twenty-one former LP were compared to 38 EP (15 with Bronchopulmonary dysplasia (BPD) [EP+], 23 without BPD [EP−]) and to 25 TC children. Peak oxygen uptake (peakV̇O_2_) was statistically lower than in the TC, but within the normal range, and without difference from the EP (LP 90.2 ± 15.1%, TC 112.4 ± 16.9%, *p* < 0.001; EP+ 97.3 ± 25.5%, EP− 85.4 ± 20.8%, *p* = 0.016 and *p* < 0.001, respectively, when compared with TC). Lung function (FEV1) was lower than normal only in the EP+ (75.6 ± 14.9% predicted, compared with 12.5 ± 87.8 in EP−, 87.5 ± 16.9 in LP and 91.0 ± 11.7 in TC). Respiratory and cardiac limitations were similar between all four study groups. Conclusions: This study demonstrated lower exercise capacity (peakV̇O_2_) in former LP children compared with healthy term children. Exercise capacity in LP was comparable to that of EP, with and without BPD. However, the exercise test parameters, specifically peakV̇O_2_, were within the normal range, and no significant physiological exercise limitations were found.

## 1. Introduction

Approximately 10% of all newborns are born preterm (before 37 weeks’ gestation) [1], of whom 75% are born in late prematurity (i.e., 34^0^ to 36^6^ weeks’ gestation) [2] and only 1% are born before 30 weeks’ gestation [3]. Complications of prematurity and their adverse consequences have been suggested to increase with the degree of prematurity but also to be present in those born in late prematurity [4,5,6].

Exercise capacity is a known predictor of morbidity and mortality in children and adults and is determined by combining the respiratory and cardiovascular systems, pulmonary oxygen uptake, and its utilization by the muscles [7].

Late preterm infants are at increased risk for morbidities involving nearly every body system when compared to term-born infants [2,8]. The data on premature babies and lung functions are contradicting; some showed reduction and some did not show a significant difference compared to term controls [9,10,11]. We are not aware of studies comparing exercise capacity using Cardio-Pulmonary-Exercise-Testing (CPET) between late and early prematurity, and term healthy children [7]. Early prematurity (<30 weeks’ gestation) and very low birthweight (<1500 g) were associated with lower physical activity, fitness, aerobic capacity, and strength compared to those of term-born control subjects [12,13,14,15,16,17]. Bronchopulmonary dysplasia (BPD) was also independently associated with poor exercise results [18,19,20]. However, there is missing data regarding the comparison between late preterms and both early preterms and/or term healthy children born at term [7,21]. Our null hypothesis was that former late preterm (LP) children will have reduced exercise capacity (peakV̇O_2_) and that exercise capacity decrease with increased degree of prematurity and with BPD diagnosis.

Our study aim was to evaluate the effect of late prematurity on exercise capacity in childhood, in comparison to early preterm (EP) (born < 30^0^ GA), and to term healthy control (TC) (>37^0^ week GA).

## 2. Methods

### 2.1. Design

In this prospective study, we compared hemodynamic, cardio-respiratory parameters and exercise capacity during Cardio-Pulmonary-Exercise-Testing (CPET) in three groups of children: those who were born in late and in early prematurity, and term controls. The study was conducted during a single visit in the CPET lab in a tertiary university-affiliated medical center. The study was approved by the Institutional Review Board of Rambam Health Care Campus (application number 0239-14-RMB), and written consent was obtained from parents of the participants; ClinicalTrials.gov Identifier: NCT04833647.

### 2.2. Subjects

The study population included three main groups: late preterm infants (born at 34^0^–36^6^ weeks’ gestation, LP), early preterm infants born <30^0^ weeks’ gestation (EP, further divided to those with BPD [EP+] and without BPD [EP−]), and term controls (TC). BPD was assessed from the computerized medical record and was defined as oxygen dependence (FiO_2_ > 21%) or high flow nasal prongs (≥2 L per minute [LPM]), continuous positive airway pressure (CPAP) or mechanical ventilation at 36 weeks’ gestation. This definition of BPD, moderate or severe (or grade 2–3), predicts early childhood morbidity [22].

The inclusion criteria were 7–10 years old and the ability to ride stationary cycle ergometer as assessed by the parents. Exclusion criteria included: immobility or severe cognitive impairment preventing following simple technician instruction, chronic lung illness (with the exception of BPD and asthma), and a congenital cardiac defect (with the exception of minimal atrial septal defect and patent foramen ovale). Children with acute febrile or upper respiratory illness in the last 2 weeks, using systemic or inhaled steroids in the previous month or inhaled bronchodilator use in the previous 24 h were excluded.

### 2.3. Participants’ Recruitment

The different infants’ groups were identified using hospital’s computerized list generated according to birth gestational age or from the pediatric pulmonary institute outpatient clinic. BPD was assessed from the computerized medical records.

An initial phone call was done between a neonatologist (O.H.) or a pediatric pulmonologist (R.B.-Y.) and the parents, explaining about the study and assessing the inclusion and exclusion criteria. Control group was recruited from healthy siblings and friends of the study group children and from healthy children visiting the children hospital with their families. At the day of the study, the parents received a repeated explanation and signed an informed consent.

### 2.4. Data Collection and Cardio-Pulmonary-Exercise-Testing (CPET)

#### 2.4.1. Anthropometric and Disease Measures

Age, sex, height, body mass, and body mass index (BMI) as well as z-scores for body mass index (BMI) based on center of disease and control (CDC) criteria were recorded [23]. Each parent completed a questionnaire regarding his child’s recent illness, medication use, lifestyle, and general information regarding habitual physical activity (see detailed supplementary: Habitual physical activity and medical background questionnaire). In addition, the Children’s Assessment of Participation and Enjoyment (CAPE) questionnaire was filled in order to measure the extent of participation in leisure, recreational, and physical activity [24]. Baseline pre-CPET pulmonary function was used for analysis of potential breathing limitation during exercise.

#### 2.4.2. Spirometry

Spirometry was performed in accordance with the European Respiratory Society and American Thoracic Society guidelines using a Quark spirometer (Cosmed, City, Italy) [25]. Spirometry measured participants forced vital capacity (FVC), forced expiratory volume in one second (FEV_1_), and maximum voluntary ventilation (MVV) were calculated using the 12-s sprint method [26]. Pulmonary function tests (PFT) were completed before the CPET and 15 min post exercise, and a fall of >12% in FEV_1_ was considered exercise induced bronchoconstriction.

#### 2.4.3. Exercise Testing with CPET

CPET was performed using a Quark CPET metabolic cart (Cosmed, Rome, Italy) according to ATS guidelines [27]. All the exercise tests were carried out by the same experienced physician (R.B.-Y.) and the same technician (M.H.) using a cycle ergometer progressive ramp protocol, beginning with no resistance warm-up lasting 1–3 min and followed with an incremental resistance adapted to the patient’s functional capacities according to the examiner’s free judgment and ranging from 5 to 20 Watts/minute. Subjects were asked to maintain a pedal speed at the desired protocol level, 60–70 rounds per minute. Gas exchange variables through a designated facemask (V2 mask, Cosmed, Rome, Italy), 12-lead ECG, blood pressure and oxygen saturation (SpO_2_) were recorded at rest, during the test and during the recovery period. SpO_2_ was measured continuously using Massimo SET 2000 (Schiller) and recorded at baseline, every 120 s, peak exercise and one, two- and five-min post exercise. Criteria for terminating the test were: inability to maintain pedaling cadence (<60 rpm), in association with subjective evidence of fatigue (sweating, hyperpnea), and one or more of the following: peakV̇O_2_ > 80% predicted [28], maximal heart ate > 80% HR predicted (HR pred = 208 − [Age × 0.7]) [29], respiratory exchange ratio (RER) > 1.0, or reaching a V̇O_2_ plateau (failure to increase oxygen uptake despite a continuous increase in work) [30]. Breathing reserve (BR) was calculated as [MVV − peakVE]/MVV and low BR was defined as BR% < 15% or BR < 11 L/min [31].

### 2.5. Statistical Analysis

Descriptive statistics in terms of mean, standard deviation, median and percentiles were calculated to the whole parameters in the study. Normal distribution of the quantitative parameters was tested by Kolmogorov–Smirnov. As a result of this test, we used ANOVA and a Kruskal–Wallis test for differences between groups with Bonferroni or Dunn’s post-tests, respectively. SPSS (SPSS Inc., Chicago, IL, USA) version 27 was used for all statistical analysis. *p* < 0.05 was considered as significant. A 10% difference in peakV̇O_2_ was considered clinically significant and, using a significance level of 5% with 80% power, a sample of 25 children in each group was required for demonstrating a statistical significant difference in peakV̇O_2_.

## 3. Results

Eighty-four children participated in the study: 21 LP, 38 EP (15 EP+, 23 EP−), and 25 TC. All patients completed PFT and maximal CPET. No adverse events were recorded (Figure 1). Demographic, anthropometric data are presented in Table 1. The control group was slightly younger (~1 year) than the study groups with no significant differences regarding height, weight, or BMI indices. As expected, LP, EP, and TC differed by birth weight and gestational age. Both LP and TC had fewer oxygen supplementation days compared to EP and no ventilation days. EP+ had more oxygen supplementation than EP− with no significant difference in ventilation days.

Pulmonary function tests are presented in Table 1 (and Appendix A). LP infants had normal spirometry values, comparable to the TC infants. EP+ had mildly reduced FEV_1_ (14.9 ± 75.6% predicted) and lower than all other groups (*p* ≤ 0.02). The FVC tended to be lower in the EP(+) group, but there was no significant difference between the groups, and all were within normal limits. None of the participants had post exercise bronchoconstriction.

CPET parameters are presented in Table 2 (and Appendix A). PeakV̇O_2_ was within the normal range and comparable between all preterm groups but statistically lower than our healthy control group (EP+ 97.3 ± 25.5%, *p* = 0.016; EP− 85.4 ± 20.8%, *p* < 0.001; LP 90.2 ± 15.1%, *p* < 0.001; compared with TC 112.4 ± 16.9%). Pre-exercise and peak SpO_2_ were preserved and similar between groups. Ventilatory equivalent for V̇CO_2_ (V̇E/V̇CO_2_ slope measured during ramp protocol up to the anaerobic threshold) was normal and similar between groups. Low BR was significantly more common in the LP group compared to TC (9/21, 43% vs. 3/24, 12%, *p* = 0.02) with no difference compared to low BR in the EP groups. Peak O_2_ pulse (V̇O_2_/HR% predicted) was comparable and within the normal range for all preterm groups) but statistically lower from the TC group.

The general questionnaires and the CAPE questionnaires did not show any significant difference in daily leisure and physical activity between the three groups.

## 4. Discussion

This study demonstrated lower exercise capacity in former LP children aged 7–10 years compared to healthy term children. Exercise capacity in LP was comparable to that of EP, with and without BPD. However, the exercise test parameters, specifically peakV̇O_2_, were within the normal range and no significant physiological exercise limitations were found. Spirometry values were also within normal range in all groups; only FEV_1_ was significantly lower in the BPD group.

In our LP group, peakV̇O_2_ was significantly lower than in the TC group, but similar to the EP groups, (with or without BPD). Svedenkrans et al. studied 218,820 young males (18–25 years) conscripting for military service who completed a modified maximal exercise test on a cycle ergometer and found lower exercise capacity in subjects born moderately preterm (32–36 weeks GA) vs. term; however, they did not report gas exchange parameters including peakV̇O_2_. Additionally, the distribution of results consisted of both LP and infants born earlier, at 32–33 weeks GA [32]. We are not aware of studies investigating the exercise capacity using Cardio-Pulmonary-Exercise-Testing (CPET) and concentrating/focusing in late prematurity [7].

Evidence from other, simpler, techniques to assess physical fitness (e.g., push-ups, handgrip strength, heart rate at the end step test, and self-rated physical activity and fitness) suggest that those born as early preterm had lower muscular fitness than controls born at term and report to engage less in leisure time physical activities [21,33]. Tamai et al. showed that children born very preterm (25–31 weeks) and moderately to late preterm (32–36 weeks) were less likely to participate in sports clubs at 7, 10, and 15 years old than children born full term (39–41 weeks) [34]. Rogers et al. compared extremely low birth weight adolescents (17 years of age) with term-born control subjects, and found lower aerobic capacity, strength, endurance, flexibility, and activity level. They concluded that these differences in fitness and physical activity are related to the interaction of effects of premature birth on the motor system together with a more inactive lifestyle [14]. Tikanmäki et al. found lower muscular fitness and less physical activity among young adults born as early preterms compared to term-born controls. However, cardiorespiratory fitness, measured by submaximal step test, was similar [20,31]. Lowe et al. also demonstrated that young adults born as preterm engaged less in leisure time physical activities than peers born at term [35]. Clemm et al. found lower participation in sport activity among children born prematurely, and demonstrated that this physical activity was positively associated with exercise capacity in preterm and term-born subjects alike [36]. Contrarily to the previous studies and similar to the current study, Welsh found no differences in physical activity between early premature infants (most with BPD and decreased FEV_1_ and V̇O_2_max) compared to the term-born group [15]. To summarize, most of these studies reported on early preterm infants, and the data were conflicting.

The precise underlying cause of lower aerobic exercise capacity in children born preterm is not entirely known, but it is probably not solely due to low breathing reserve (e.g., relatively preserved lung function as reflected by spirometry). It could be affected by a number of interrelated parameters including mechanical ventilatory constraints, impaired cardiovascular function and altered muscle bioenergetics [7], as well as lower physical activity among children born prematurely [36].

In our study, intriguingly, respiratory limitation (e.g., low BR) was more common in the LP group (43%) compared to TC (12%), with no statistical difference from other study groups (EP+ 33%, EP− 26%). While this potential limitation could be explained by low pulmonary function tests in the EP+ group, we have not found an explanation in the other groups. Furthermore, the ventilatory equivalents (V̇E/V̇CO_2_ slope, a submaximal marker for increased ventilatory drive that is related to the amount and sensitivity of central chemoreceptors and the ventilatory dead-space) were similar between all groups and could not further explain our above findings.

Although peakV̇O_2_ was comparable between all preterm infants (regardless of the prematurity level or BPD status) and significantly lower than in the TC group, lung function was significantly lower only in the BPD group. Similar to the results in our study, the diagnosis of BPD in a preterm infant was shown to be associated with impaired lung function later in life [15,37]. However, regardless of BPD status, early prematurity can result in impaired lung function in the long term. Both Kotecha et al. and Edwards et al., in two different systemic reviews, suggest that all early preterm-born survivors are at risk for long-term deficits in FEV_1_ [38,39]. The two aforementioned systemic reviews related mainly to the “old” BPD era. Kaplan et al. studied 53 neurologically intact early preterm children and adolescents (28 with “new” BPD and 25 without BPD) and reported FEV_1_ within the normal range (82% pred) but lower than control [40,41]. On the contrary, other studies failed to show difference in spirometry and wheezing between term-born and late preterm infants during childhood, adolescence, and adulthood [10,42,43].

In addition to the risk of long-term deficits in lung function, prematurity (mainly EP) is also associated with cardiovascular changes including alterations in arterial distensibility [44], smaller right ventricular size and greater mass [45], ejection fractions below the lower limit [45], and autonomic dysfunction [17], all which potentially can influence exercise capacity. Reduced aortic intima-media thickness (IMT) and reduced heart rate variability (autonomic function) were shown in infants born late preterm, compared to term infants during infancy and adulthood [46,47]. Yang et al. showed that, compared to term controls, preterms born at 26–30 weeks of gestation have reduced physical activity, impaired lung, altered left ventricular structure/function, and reduced right atrial/ventricular size. Adjustment for BMI, lung function, and cardiac structure/function explained most of the exercise group-differences [48].

In our study, O_2_pulse parameters (e.g., low value of V̇O_2_/HR or flattening of the V̇O_2_/HR curve are considered markers for decreased cardiac stroke volume) were similar and within the normal range for all study groups and only peak O_2_pulse was statistically lower from the TC group. O_2_ pulse curve flattening was not observed in any of the patients. Baseline and peak exercise oxygen saturation (Sat. O_2_) were normal in all participants. Baseline echocardiography was not performed in this study, but no significant heart abnormalities were reported. These findings could result from possible lower habitual physical activity in the study groups compared to the control group.

Another possible functional limitation could be the cognitive system and motor performance. In a previously mentioned study, Svedenkrans et al. showed lower cognitive function for young men born moderately preterm (32–36 weeks GA) with a positive correlation between cognition and exercise capacity [32]. Pikel et al. studied 178 preterms (8–18 years) using field tests and concluded that EP as well as moderate preterms (32–36 weeks GA) exert longitudinal effects on exercise capacity and motor performance up to early adulthood [49]. In the current study, we did not evaluate these dimensions.

The strength of our study is the comparison of LP to both early preterm groups (with and without BPD) and healthy controls and using the same comprehensive tests (e.g., CPET), investigators, technicians, and equipment. The control group was chosen mostly from the same surrounding of the preterm infants (usually siblings and classmates). Another advantage is that it was done on children born as preterm in the “new BPD” era, while many previous studies assessed preterm infants with “old BPD”, thus less applicable to our time.

Interestingly, although even the low degree of prematurity of the LP group had a significant influence on exercise capacity compared to TC, we found no significant difference between LP and EP groups. This finding is intriguing. Theoretically, all the above-mentioned explanations for lower exercise capacity (e.g., habitual physical activity, lung function, cardiac function, muscle bioenergetics) may have an effect towards a further decrease in exercise capacity as gestational age at birth decreases. However, the questionnaires in our study failed to describe any difference regarding physical activity between the LP and EP groups. Moreover, it was observed that lung function alone is not necessarily a major factor influencing exercise capacity [39]. In the context of growth and thriving, there was no difference between the groups in terms of weight, height, and BMI. As mentioned above, individuals born preterm can have structural cardiac remodeling and altered cardiac growth and function later in life [50]. However, we did not find in the literature a detailed comparison regarding cardiac function between early and late preterms. In our study, we did not perform an echocardiography, so we are unable to determine if this was a main factor influencing our results. We hope that our research will stimulate future studies focusing on possible mechanisms for this phenomenon.

Our study has several limitations. First, it is a relatively small sample size. This could result in beta error. Our relatively small number of participants was partially due to the challenge of voluntary recruitment in the COVID-19 era. The exclusion criteria of children who were unable to participate in the CPET (including respiratory or neurological deficits) could create a selection bias. The participants did not have routine baseline echocardiography and validated physical activity questionnaire. The mean VO_2_% predicted in our control group was 112.4 ± 16.9%. This may reflect our pediatric population characteristics; however, a selective bias of fit children cannot be excluded.

In conclusion, our study demonstrated lower exercise capacity in former LP children aged 7–10 years compared to healthy term children and comparable to that of EP (with and without BPD). The PFTs in LP were within the normal range in our study and could not explain the lower exercise capacity. The etiology for lower exercise capacity in LP is unclear and may include reduced physical activity, low breathing reserve, altered muscle metabolism, and altered heart structure and function. However, the exercise test parameters, specifically peakV̇O_2_, were within the normal range both in EP and LP, and no significant physiological exercise limitation was found. These results are encouraging. Nevertheless, a personalized respiratory, fitness, and physical activity follow-up in cases of prematurity should be encouraged, and parents of all preterm infants should actively encourage their children to participate in regular childhood physical activities and sports. Longer and larger prospective case-control studies are needed focusing on the physiological mechanisms to possible exercise limitations.

## Figures and Tables

**Figure 1 jpm-12-01547-f001:**
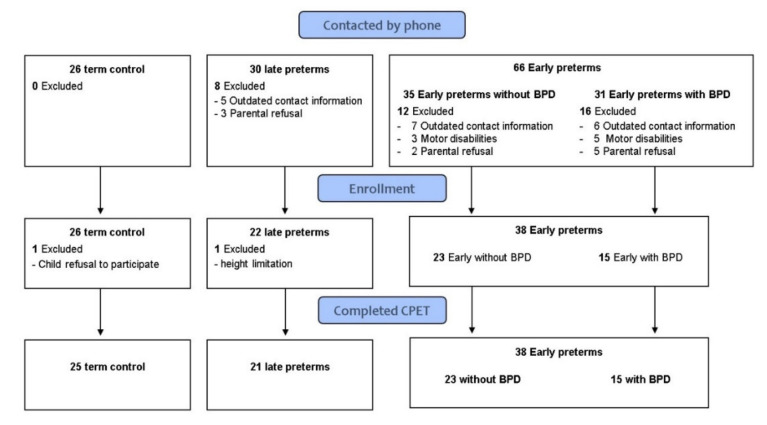
Flow chart of participants in the study.

**Table 1 jpm-12-01547-t001:** Demographic, anthropometric and lung function data (term, early preterm, late preterm).

	GA = 34–36.6 Weeks (Late Preterm) *n* = 21 1	GA ≤ 30 Weeks (Early Preterm) *n* = 38 2	Healthy Control Born at Term *n* = 25 3	*p*-Value *
Age (years)	9.94 ± 0.87	9.63 ± 1.20	8.84 ± 0.93	*p*^2^ < 0.001 *p* ^3^ = 0.014
Male (%)	12 (57%)	19 (50%)	13 (52%)	*p* = 0.87
Gestational Age (weeks)	34.9 ± 1.05	28.44 ± 1.5	39.5 ± 1.4	*p*^1,2,3^ < 0.0001
Birth Weight (gr)	2373 ± 473	1105 ± 280	3315 ± 469	*p*^1,2,3^ < 0.0001
Oxygen Supplementation (days) median 25–75	0 [0–1]	38 [8.5–67.50]	0 [0–0]	*p*^1,3^ < 0.001
Ventilation (days) median (25–75 quartile)	0 [0–0]	6 [3–30]	0 [0–0]	*p*^1,3^ < 0.001
Height (cm)	137.4 ± 7.7	134.6 ± 9.7	133.3 ± 6.5	*p* = 0.25
Weight (kg)	32.6 ± 6.4	33.2 ± 12.9	30.9 ± 7.6	*p* = 0.66
BMI	17.1 ± 2.2	17.9 ± 4.5	17.2 ± 2.8	*p* = 0.62
BMI percentiles	49.5 ± 29.3	53.7 ± 32.7	55.5 ± 31.5	*p* = 0.81
BMI z score median (25–75)	0.12 [(−0.84)–(0.73)]	0.11 [(−0.81)–(0.93)]	0.28 [(−0.55)–(1.09)]	*p* = 0.75
FEV_1_ (L/s)	1.73 ± 0.38	1.57 ± 0.38	1.65 ± 0.32	*p* = 0.28
FEV_1_ (% predicted)	87.5 ± 16.9	82.9 ± 14.6	91.04 ± 11.7	*p* = 0.095
FVC (L)	2.02 ± 0.4	1.87 ± 0.42	1.89 ± 0.37	*p* = 0.17
FVC (% predicted)	94.09 ± 13.6	88.9 ± 13.04	95.5 ± 11.6	*p* = 0.11

* *p*
^1^ = 1 vs. 2; *p*
^2^ = 1 vs. 3; *p*
^3^ = 2 vs. 3. GA—Gestational Age; W—weeks; BPD—bronchopulmonary dysplasia; BMI = body mass index; FEV1 = Forced expiratory volume in one second.

**Table 2 jpm-12-01547-t002:** CPET parameters (term, early preterm, late preterm).

	GA = 34–36.6 Weeks (Late Preterm) *n* = 21 1	GA ≤ 30 Weeks (Early Preterm) *n* = 38 2	Healthy Control Born at Term *n* = 25 3	*p*-Value *
PeakV̇O_2_ Absolute (mL/min)	1206 ± 248	1146 ± 348	1380 ± 260	*p*^2^ = 0.02 *p* ^3^ = 0.014
PeakV̇O_2_ Specific (mL/kg/min)	37.6 ± 6.8	36.4 ± 11.4	45.2 ± 7.4	*p* ^2^ = 0.038 *p* ^3^ = 0.002
PeakV̇O_2_ (%Pred)	90.2 ± 15.1	90.08 ± 23.19	112.4 ± 16.9	*p* ^2,3^ < 0.001
Peak HR (bpm)	193.4 ± 7	190.08 ± 10.65	191.8 ± 9.8	*p* = 0.46
Peak HR (%pred)	96.1 ± 3.9	94.4 ± 5.39	95.0 ± 5.0	*p* = 0.44
V̇E/V̇CO_2_ Slope	34.4 ± 5.6	35.91 ± 6.36	34.4 ± 4.3	*p* = 0.52
Peak O_2_ pulse (%pred)	94.1 ± 15.4	95.5 ± 23.01	118.8 ± 19.1	*p* ^2,3^ < 0.001
Peak O_2_ pulse (V̇O_2_/HR)	6.2 ± 1.1	6.0 ± 1.71	7.2 ± 1.4	*p* ^2^ = 0.031 *p* ^3^ = 0.008
Sat. O_2_ Pre	98.7 ± 1.05	99.03 ± 0.91	99.1 ± 1.0	*p* = 0.40
Sat. O_2_ Post	98.5 ± 1.4	98.71 ± 1.31	99.1 ± 1.1	*p* = 0.27
Peak V̇E (L/min)	48.8 ± 12.4	42.4 ± 11.78	47.5 ± 9.3	*p* = 0.076
BR (L)	20.4 ± 11.0	19.77 ± 13.38	20.6 ± 12.0	*p* = 0.97
BR (%)	28.7 ± 13.5	30.18 ± 17.61	29.3 ± 12.9	*p* = 0.94
Breathing limitation Low Normal	9 (43%) 12 (57%)	11 (29%) 27 (71%)	3 (12%) 21 (88%)	*p* ^2^ = 0.02

* *p*
^2^ = 1 vs. 3; *p*
^3^ = 2 vs. 3. GA—Gestational Age; W—weeks; BPD—bronchopulmonary dysplasia; PeakV̇O_2_—oxygen uptake at the peak of exercise; HR—heart rate; V̇E—minute ventilation; V̇CO_2_—carbon dioxide production; Sat—Saturation; BR—breathing reserve.

## Data Availability

The data that support the findings of this study are available from the corresponding author (L.B.) upon reasonable request.

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
