# Peer review of "Cardiopulmonary Exercise Testing in Childhood in Late Preterms: Comparison to Early Preterms and Term-Born Controls"

_jpm, 2022, doi:10.3390/jpm12101547_

Round 1

Reviewer 1 Report

This is a very well-written and clear manuscript entitled “Evaluating the effect of late prematurity on cardiopulmonary exercise testing in childhood.” I appreciate the ease of reading, the thoroughness in the introduction and conclusion, and the direct result reporting (including figure 1 with recruitment information). My comments are small:

-          The authors collected data on gender, BMI, and lifestyle/physical activity. Is there space to run statistics including these variables? Does background literature inform these thoughts? Your discussion very appropriately talks about the complex factors involved in capacity, and it seems these variables would be of interest.

-          I worry that the conclusion in the abstract seems a bit overstated. The results reviewed in the abstract do not seem to support such a statement unless significant in results.

Author Response

Dear reviewers,

We would like to thank you for your valuable time and significant comments.

Please find below our point to point responses.

All additions to the article are marked in italic.

Reviewer Report (Round 1):

Reviewer 1

We thank the reviewer for hers/his important and helpful comments for improving our article.

  1. The reviewer note: “The authors collected data on gender, BMI, and lifestyle/physical activity. Is there space to run statistics including these variables? Does background literature inform these thoughts? Your discussion very appropriately talks about the complex factors involved in capacity, and it seems these variables would be of interest.”

We agree with this comment and made the following changes:

  • We elaborated on the part of the questionnaires and added in the “Methods” (under 2.4.1. Anthropometric and Disease Measures, page 3)

In addition, the Children’s Assessment of Participation and Enjoyment (CAPE) was filled in order to measure the extent of participation in leisure, recreational and physical activity. [25]

  • We added at the end of the “Result” section: The general questionnaires and the CAPE questionnaires didn’t show any significant difference in daily leisure and physical activity between the 3 groups.” (Page 8)
  • In Table 1, now gender, height, weight, BMI, BMI percentiles and Z scores are presented and compared between the 3 groups (showing no statistical difference).
  • In the discussion we added : “Contrarily to the previous studies and similar to the current study, Welsh found no differences in physical activity between early premature infants” (1st paragraph page 8)

  1. The reviewer note: “I worry that the conclusion in the abstract seems a bit overstated. The results reviewed in the abstract do not seem to support such a statement unless significant in results.”

We thank the reviewer for this comment and made the following changes:

  • We changed the “result” part in the abstract that now states:

Results: Eighty-four children (age 9.6±1.0 years, 48% girls) participated. Twenty-one former LP were compared to 38 EP (15 with Bronchopulmonary dysplasia (BPD) [EP+], 23 without BPD [EP-]) and to 25 TC children. Peak oxygen uptake (peakV̇O2) was statistically lower than in the TC, but within the normal range, and without difference from the EP (LP 90.2±15.1%, TC 112.4±16.9%, p<0.001; EP+ 97.3±25.5%, EP- 85.4±20.8%, p=0.016 and p<0.001, respectively, when compared with TC). Lung function (FEV1) was lower than normal only in the EP+ (75.6±14.9% predicted, compared with 87.8±12.5 in EP-, 87.5±16.9 in LP and 91.0±11.7 in TC). Respiratory and cardiac limitations were similar between all four study groups.”

We hope that this change makes the results clearer and therefore also support our conclusion, based on the Peak oxygen uptake (peakV̇O2) results.

Reviewer 2 Report

Dear Authors.

The article you have submitted for review must be considerably improved for its publication to achieve the impact and interest in the scientific community that the data presented could have.

On reading the manuscript, from the title to the introduction, we can see that there is no reason to add the early preterm (EP) group to the work since both the title and the introduction preferentially mention the late preterm (LP) group. The only mention of EPs is 4 bibliographic references (numbers 4 to 6) where it is the LP group that is analyzed. The data on EPs are very interesting but little used in the final discussion of the manuscript. I suggest that if you want to keep these data, include them in the title, and give more background information to support their comparison not only with the control group, but also with the LP, which you do, but you discuss it very poorly, finally losing one of the great potentialities of the work submitted by you.  The statistics presented by you do not mention the type of post-test that you perform after the ANOVA, I think I suppose it is a Bonferroni or a Newmman-Keuls, please clarify this point.

The results were well presented, describing the tables and main findings in a very good way.

The discussion points in a very good way to the most salient results of the LP group compared to the control group, however, only timid mention is made of the results of the EP group and their comparisons with the LP group. As I mentioned before the LP vs EP comparison was very interesting to explore, to see how those weeks or days of difference might impact the subsequent life of the individual, this idea was ultimately not addressed. 

I urge the authors to improve the writing and analysis done to publish these fascinating data.

Author Response

Dear reviewer,

We would like to thank you for your valuable time and significant comments.

Please find below our point to point responses.

All additions to the article are marked in italic.

Reviewer Report (Round 1):

Reviewer 2

We thank the reviewer for hers/his important and helpful comments for improving our article.

The reviewers’ notes:

  1. On reading the manuscript, from the title to the introduction, we can see that there is no reason to add the early preterm (EP) group to the work since both the title and the introduction preferentially mention the late preterm (LP) group. The only mention of EPs is 4 bibliographic references (numbers 4 to 6) where it is the LP group that is analysed. The data on EPs are very interesting but little used in the final discussion of the manuscript. I suggest that if you want to keep these data, include them in the title, and give more background information to support their comparison not only with the control group, but also with the LP, which you do, but you discuss it very poorly, finally losing one of the great potentialities of the work submitted by you

We thank the reviewer very much for this important comment. We feel that it is important and essential to add the EP group to our study. We believe that comparing LP to both TC and EP gives a more thorough data regarding the effect of prematurity and its degree on exercise capacity, our primary outcome. Once we have demonstrated that peakVO2 was significantly lower in LP compared to TC, it is also important to show and compare this result to the EP group (that was studied more in the literature), as well. In both theoretical cases, if the EP’s peakVO2 was significantly lower or comparable (as we found), this is an interesting and important information as well as the discussion that follows.

Based on the reviewer comment we made the following changes to our manuscript:

  • Title was changed to: “Cardiopulmonary Exercise Testing in Childhood in Late Preterms: Comparison to Early Preterms and Term-Born Controls.
  • In the abstarct we repharsed the aim ( similar to the introduction ) :We aim to assessthe effect of late prematurity on exercise capacity in childhood and to compare it to early preterm (EP) (born <300 GA), and to term healthy control (TC) (>370 week GA).

  • In the introduction (including 9 references (13-21) relevant to EP and BPD): Early prematurity (<30 weeks' gestation) and very low birthweight (<1500g) were associated with lower physical activity, fitness, aerobic capacity and strength compared to those of term-born control subjects.[13–18] Bronchopulmonary dysplasia (BPD) was also independently associated with poor exercise results.[19–21]. However, there is missing data regarding the comparison between late preterms and both early preterms and/or term healthy children born at term.[7,22]”.

As mentioned, we didn’t find in the literature comparisons between EP and LP. Therfore , we didn’t elaborate on such comparisons made in the past in the introduction part. We think it is important to include EP in our manuscript. Moreover, now references 13-21 are dealing with different aspects of exercise and physical activity in EP, and in BPD which is related to early prematurity as well.

  • In the discussion we have made some changes. Over all the EP group is mentioned with references to different theoretical etiologies regarding the influence of early prematurity and exercise activity.

We added a new paragraph in the discussion part:

Interestingly, although even the low degree of prematurity of the LP group had a significant influence on exercise capacity compared to TC, we found no significant difference between LP and EP groups. This finding is intriguing. Theoretically, all the above-mentioned explanations for lower exercise capacity (e.g., habitual physical activity, lung function, cardiac function, muscle bioenergetics) may have an effect towards a further decrease in exercise capacity as gestational age at birth decreases. However, the questionnaires in our study failed to describe any difference regarding physical activity between the LP and EP groups. Moreover, it was observed that lung function alone is not necessarily a major factor influencing exercise capacity.[43] In the context of growth and thriving, there was no difference between the groups in terms of weight, height and BMI. As mentioned above, Individuals born preterm can have structural cardiac remodeling and altered cardiac growth and function later in life.[56] However, we did not find in the literature a detailed comparison regarding cardiac function between early and late preterms. In our study we did not perform an echocardiography, so we are unable to determine if this was a main factor influencing our results. We hope that our research will stimulate future studies focusing on possible mechanisms for this phenomenon.

  1. The statistics presented by you do not mention the type of post-test that you perform after the ANOVA, I think I suppose it is a Bonferroni or a Newmman-Keuls, please clarify this point

We agree and thank the reviewer for this comment. We added in the “methods” à “statistical analysis” (part 2.5 in methods) the following sentence:

Normal distribution of the quantitative parameters was tested by Kolmogorov-Smirnov. As a result of this test, we used Anova and Kruskal Wallis test for differences between groups with Bonferroni or Dunn’s post-tests, respectively

  1. The discussion points in a very good way to the most salient results of the LP group compared to the control group, however, only timid mention is made of the results of the EP group and their comparisons with the LP group. As I mentioned before the LP vs EP comparison was very interesting to explore, to see how those weeks or days of difference might impact the subsequent life of the individual, this idea was ultimately not addressed.

As noted above, we thank the reviewer for this comment, agreed to it, and addressed it in our 1st response above.

Round 2

Reviewer 2 Report

Dear Authors.

I agree with the new version. Congratulations